# Au_30_(P*^i^*Pr_2_*^n^*Bu)_12_Cl_6_—An Open Cluster Provides Insight into the Influence of the Sterical Demand of the Phosphine Ligand in the Formation of Metalloid Gold Clusters

**DOI:** 10.3390/molecules29020286

**Published:** 2024-01-05

**Authors:** Markus Strienz, Andreas Schnepf

**Affiliations:** Institute of Inorganic Chemistry, Universität Tübingen, Auf der Morgenstelle 18, 72076 Tübingen, Germany; markus.strienz@uni-tuebingen.de

**Keywords:** gold cluster, cluster formation, phosphine ligand, X-ray structure, cluster transformation, steric effect

## Abstract

Phosphine-stabilized gold clusters are an important subgroup of metalloid gold cluster compounds and are important model compounds for nanoparticles. Although there are numerous gold clusters with different phosphine ligands, the effect of phosphine on cluster formation and structure remains unclear. While the linear alkyl-substituted phosphine gold chlorides result in a Au_32_ cluster, the bulky *^t^*Bu_3_P leads to a Au_20_ cluster. The reduction of (*^i^*Pr_2_*^n^*BuP)AuCl, with the steric demand of the phosphine ligand between the mentioned phosphines, results in the successful synthesis and crystallization of a new metalloid gold cluster, Au_30_(P*^i^*Pr_2_*^n^*Bu)_12_Cl_6_. Its structure is similar to the Au_32_ cluster but with two missing AuCl units. The UV/Vis studies and quantum chemical calculations show the similarities between the two clusters and the influence of the phosphine ligand in the synthesis of metalloid gold clusters.

## 1. Introduction

Nanoparticles are localized at the border between the molecular and the solid states [1,2]. As a result, they can exhibit properties of both molecules and solids, as well as novel properties not found in either. These properties are dependent on size, shape, material, etc. This makes nanoparticles interesting compounds with applications in catalysis, medicine, electronics, etc. [3,4,5]. Accessing and investigating nanoparticles is challenging, as they represent a wide size range and, hence, show different properties. One way to investigate the size–property relationship is with cluster compounds due to their atomic precision and, therefore, well-defined properties [6]. This atomic precision allows cluster compounds to be crystallized, providing knowledge of the exact location of each atom. This distinguishes cluster compounds from nanoparticles.

The first gold cluster Au_11_(PPh_3_)_7_(SCN)_3_, which is a metalloid gold cluster, was synthesized and crystallized by Malatesta in 1969 through the reduction of (PPh_3_)AuSCN with NaBH_4_ in ethanol [7]. This gold cluster consists of 11 gold atoms with an average oxidation state of the gold atoms between 0 and +1, and it can thus be seen as an intermediate between solid Au(0) and gold chlorides Au(I/III). The arrangement of the gold atoms in the Au_11_ cluster is referred to as a centaur polyhedra, being a combination of an icosahedron and a cube. While the term ‘metal cluster’ was defined by Cotton as a molecule with at least one metal–metal bond, the term ‘metalloid’ can be employed to emphasize the proximity of such clusters to the metallic solid state when at least one naked gold atom is present in the cluster and if there are more metal–metal than metal–ligand bonds [8,9]. This is in contrast to smaller clusters like [Au_6_(PPh_3_)_6_][NO_3_]_2_, in which gold atoms are arranged in the form of two edge-sharing tetrahedra, with each gold atom being bound to a phosphine ligand. Consequently, there is no naked gold atom, and they have a more molecular-like configuration [10]. The reduction of various gold compounds, mostly stabilized by aryl phosphines, produced a series of small gold cluster compounds, the largest of which was a [Au_39_(PPh_3_)_39_Cl_6_]Cl_2_ cluster synthesized in 1991 [11]. This changed with the reduction of thiol-stabilized gold precursors and the synthesis of the Au_102_(*p-*MBA)_44_ cluster, the first metalloid gold cluster with more than 100 gold atoms. The Au_279_(SPh*^t^*Bu)_84_ cluster, which is the largest structurally characterized metalloid gold cluster to date, is another thiolate-stabilized cluster [12,13]. In addition to phosphine and thiolates, alkynyls and NHCs have been used as ligands for the stabilization of gold clusters [14,15]. There are also gold clusters with other metal core atoms. These atoms can be from either the main group or transition group metals, and may be introduced during synthesis or later introduced into the gold cluster [16,17]. In recent years, a wide range of clusters with different shapes and ligands have been synthesized, utilizing various parameters [18,19,20,21].

Nonetheless, it is not actually possible to foresee the result of a reaction before the reduction and without crystallization. In reaction systems in which numerous bonds are formed and broken within a short timescale, even minor adjustments during the synthesis can influence the outcome of the reaction. For example, the reduction of (Ph_3_P)AuCl with NaBH_4_ yields a Au_11_(PPh_3_)_7_Cl_3_ cluster, isostructural to the Au_11_(PPh_3_)_7_(SCN)_3_ cluster, whereas the reduction of (Ph_3_P)AuNO_3_ yields a Au_9_(PPh_3_)_8_(NO_3_)_3_ cluster [22,23]. The Au_9_(PPh_3_)_8_(NO_3_)_3_ cluster form can be best described by two bicapped pyramids, sharing one gold atom. Several important parameters, such as the stabilizing ligand, solvent, work-up procedure, and reducing agent, are critical factors that need consideration, in addition to the anion.

While slight changes in the phosphine ligand (e.g., the use of P(C_6_H_4_Cl)_3_ instead of P(C_6_H_5_)_3_) still yields a Au_11_ cluster, the reduction of alkyl-substituted phosphine gold chloride like (Et_3_P)AuCl results in clusters of the composition Au_32_(PR_3_)_12_Cl_8_ [24,25]. These clusters, where R can be ethyl, *n*-propyl, or *n*-butyl, consists of an icosahedral Au_12_ core surrounded by 20 gold atoms in the form of a pentagon dodecahedron. However, there are differences among these three clusters. The yield of the Au_32_ clusters increases as the length of the alkyl chain increases, from 7% for Et_3_P to 14% for nBu_3_P. In a similar manner, the solubility of these three clusters increases with the ligand length. When stabilized with nBu_3_P, the cluster is soluble in almost every organic polar and nonpolar solvent, including pentane. In contrast, the Au_32_ cluster stabilized with Et_3_P requires strong polar solvents, such as DCM.

The influence of the ligand on physical properties, like absorption and luminescence, has been studied previously. Jin et al. demonstrated that the fluorescence quantum yield of three Au_25_(SR)_18_ clusters with different thiolate ligands is directly influenced by the ligands’ ability to donate electron density to the metal core [26]. Therefore, the selenoate-stabilized Au_25_(SePh)_18_ cluster exhibits a lower quantum yield compared to the thiolate-stabilized Au_25_(SR)_18_ cluster [27]. The absorption spectra of the clusters strongly correlate with the ligand type, whereby the UV/Vis spectra of a Au_36_ and a Au_44_ cluster are completely different for the alkynyl-stabilized clusters [28]. However, small changes in the ligand, such as the three phosphines that stabilize the Au_32_ clusters, do not cause significant shifts in the absorption spectra.

While the Au_32_(PEt_3_)_12_Cl_8_ cluster crystallizes from the THF extract, layering with pentane, extraction with DCM, and layering with Et_2_O result in the crystallization of a Au_54_(PEt_3_)_18_Cl_12_ cluster [29]. The structure of the Au_54_ cluster is related to that of the Au_32_ cluster and can be considered as two merged Au_32_ clusters (vide infra). It is worth noting that the Au_54_ cluster was only detected when using the PEt_3_ ligand and not for the phosphines with the longer alkyl chain. Thus, it is plausible that the formation of a certain cluster is influenced by the steric demand of the phosphine ligand. 

By using the sterically challenging P*^t^*Bu_3_ ligand instead of one of the aforementioned phosphines, a Au_20_(P*^t^*Bu_3_)_8_ cluster is formed [30]. The structure of Au_20_ differs significantly from that of Au_32_ and Au_54_, as the shape of Au_20_ is a central hollow cuboctahedron with eight gold atoms covering each of the eight triangular faces of the cuboctahedron. Additionally, the gold atoms in Au_20_ are in the 0 oxidation state, while Au_32_ contains 8 and Au_54_ has 12 gold atoms in the +1 oxidation state, which is attributed to the presence of 8 or 12 chloride substituents, respectively. The Au_20_ together with an Au_22_ cluster represent the only examples of gold clusters in which all gold atoms have an oxidation state of 0 [31]. To investigate the influence of the steric properties of the phosphine on cluster formation in more detail, we synthesized different substituted phosphines and used them in cluster synthesis. Phosphines are well suited for this because the three organic substituents can be changed separately, allowing for small changes in the different phosphine properties. Using P*^i^*Pr_2_*^n^*Bu, with steric demands positioned between P*^n^*Bu_3_ and P*^t^*Bu_3_, we were unable to synthesize a Au_32_, Au_54_, or Au_20_ cluster. However, we discovered a new Au_30_ cluster compound, akin to Au_32_, which is presented in the following.

## 2. Results and Discussion

The (*^i^*Pr_2_*^n^*BuP)AuCl **2** was dissolved in ethanol and reduced with NaBH_4_. The initially colorless solution rapidly turned to black, and, after one hour, the solvent was removed. The crude black solid was extracted with cyclopentane and stored at a temperature of 6 °C. The Au_30_(P*^i^*Pr_2_*^n^*Bu)_12_Cl_6_
**1** crystallizes after a few weeks as black rectangular crystals in the P2_1_/n space group (Figure 1a). On the basis of the crystal structure, the size of the gold core is 0.9 nm, with a ligand shell of 0.8 nm, resulting in a total size of 1.7 nm. It is also possible to crystallize **1** using pentane and hexane for the extraction and storing the extract at a temperature of −30 °C, however, leading to crystals of lower quality.

The gold core of **1** is composed of 30 gold atoms, with 12 gold atoms forming a hollow icosahedral inner shell (Figure 1b,d). The interatomic distances between these gold atoms measure 284.9 ± 5.7 pm, which is marginally shorter than the distances between the gold atoms in the solid state with 288 pm. The remaining 18 gold atoms form a pentagonal dodecahedron with two opposite corners missing, and the distance of the gold atoms between the inner and outer shell is 278.4 ± 5.7 pm (Figure 1c). In comparison to the distances found in the inner shell, the outer shell of compound **1** exhibits longer distances between the gold atoms, measuring 314.8 ± 17.3 pm. Incomplete geometric structures are well known from other gold cluster compounds. The Au_133_(SPh*^t^*Bu)_52_ cluster is built from two icosahedral Au_13_ and Au_43_ shells, encapsulated by a Au_52_ rhomboicosahedron with eight atoms missing [32]. The core of the Au_144_(SCH_2_Ph)_60_ cluster is similar to that of the Au_133_ cluster and differs only because of the absence of the central gold atom and the presence of an outer Au_60_ rhombohedral shell [33]. As such, this Au_144_ cluster is the geometrically closed cluster of the Au_133_. 

The geometrically closed equivalents of **1** are the Au_32_ clusters. The inner shell of the Au_32_ cluster is also a hollow Au_12_ icosahedron (Figure 2a), but the outer shell differs from **1** by two additional gold atoms, resulting in a Au_20_ pentagonal dodecahedron (Figure 2b). Three different Au_32_ clusters are known and can be synthesized from the reduction of phosphine-stabilized gold chlorides, (R_3_P)AuCl, in which the phosphine ligands are substituted with linear alkyls (R = Et, *^n^*Pr, *^n^*Bu) [25].

These phosphine ligands are sterically less demanding compared with P(*^i^*Pr_2_*^n^*Bu) (this assumption follows from the fact that both the measured and calculated Tolman cone angles for (*^i^*Pr_3_P)Au exceed those for (*^n^*Bu_3_P)Au, and, therefore, the cone angle for (*^i^*Pr_2_*^n^*BuP)Au should be larger than for phosphines, substituted with linear alkyls) [34].

By comparing the Au-Au distances of **1** to those found in Au_32_(P*^n^*Bu_3_)_12_Cl_8_
**3** (the gold–gold distances differ only slightly for the different Au_32_ clusters), the Au-Au distances within the icosahedral core of **3** are 285.1 ± 3.3 pm, similar to the distances in **1** (Table 1).

However, a noticeable difference in distances is seen within the bonds connecting the inner and outer shell of the structure, as well as within the outer shell itself. The distances between the gold atoms in the outer shell are 312.6 ± 11.7 pm within **3**, which is over 2 pm shorter when compared with **1**. The larger standard deviation of 17.3 pm is even more noteworthy; this results from the decrease in symmetry in compound **1**.

Six out of the eighteen gold atoms in the outer shell of **1** bind to chlorine atoms with an average Au-Cl bond length of 239.9 ± 0.3 pm. Compared with the educt (*^i^*Pr_2_*^n^*BuP)AuCl **2**, these bonds are almost 10 pm longer. The chlorine atoms are arranged in a cube shape, with two diagonally opposite corners absent. The Au_32_ cluster contains these two missing AuCl units from **1**, with the chlorine atoms arranged in a cubic formation.

The twelve remaining gold atoms in **1** are coordinated by a *^i^*Pr_2_*^n^*BuP ligand with an average gold–phosphorus bond length of 229.4 ± 1.1 pm. In comparison, **2** has a shorter bond length by approximately 5 pm, at 224.1 pm. The phosphorus atoms in **1** form an icosahedron with two different chemical surroundings for the phosphorus atoms. Six of the twelve phosphorus atoms are in the vicinity of one chloride and one phosphine ligand, while the remaining six phosphines are surrounded by two chlorides and one phosphine ligand (Appendix A). This leads to two signals in the ^31^P NMR spectrum of **1** at 81.38 and 85.69 ppm with identical intensities (Figure 3). An additional signal with an intensity of around 2.5% can be observed at a shift of 120.23 ppm. An explanation for this additional small signal can be found in the crystal structure solution. Hence, in addition to the 30 gold atoms of **3**, an electron density remains at the two missing corners of the pentagonal dodecahedron. This electron density could be refined to 3% of a gold atom. This result indicates that approximately 3% of the clusters in the crystal consist of 32 gold atoms. However, it remains unclear if the two gold atoms are bound to a chlorine atom, as the respective refinement of a 3% chlorine atom site position was not possible, and, thus, it is uncertain whether the second cluster compound is Au_32_(P*^i^*Pr_2_*^n^*Bu)_12_Cl_6_ or Au_32_(P*^i^*Pr_2_*^n^*Bu)_12_Cl_8_. This issue could also not be clarified by mass spectrometry, as the mass spectrometric investigations on dissolved crystals of **1** failed.

Reducing (Et_3_P)AuCl with NaBH_4_ and layering the resulting THF extract with pentane yields the previously mentioned Au_32_(PEt_3_)_12_Cl_8_ cluster. On the other hand, extracting the solid with CH_2_Cl_2_ and layering it with Et_2_O leads to the formation of the Au_54_(PEt_3_)_18_Cl_12_
**4** cluster [29].
(Et_3_P)AuCl + NaBH_4_ in ethanol, extr. THF/pentane → Au_32_(PEt_3_)_12_Cl_8_
(Et_3_P)AuCl + NaBH_4_ in ethanol, extr. CH_2_Cl_2_/Et_2_O → Au_54_(PEt_3_)_18_Cl_12_

Cluster **4** is composed of three shells. The inner shell, which is a Au_2_ dumbbell, is surrounded by 22 gold atoms (Figure 2c). The arrangement of these 22 gold atoms can be described as two icosahedra that have been sliced and fused together. The Au-Au distances are within 284.8 ± 10.1 pm close to the distances in **1** and **3**. The gold Au_2_ dumbbell is located at the central position of each icosahedron and, therefore, differs from the hollow inner shells of **1** and **3**. The outer shell of **4** consists of two merged truncated pentagonal Au_15_ dodecahedra with an average gold–gold distance of 306.0 ± 12.5 pm. This is more than 6 pm shorter compared to **3** and **1**, whereby the gold bonds between both shells in **4** exceed the bonds in **1** and **3** with 281.5 ± 9.4 pm (Figure 2d). The structure of **4** resembles the composition of two fused Au_32_ cluster. Although direct evidence is missing, it is very likely that the phosphine ligand PEt_3_ is essential for the formation of **4**, since no formation of a Au_54_ cluster, similar to **4**, was ever observed for the ligands P*^n^*Pr_3_ and P*^n^*Bu_3_. It is assumed that there is an undefined, uncharacterized precursor in solution after the reduction of (Et_3_P)AuCl and that the formation of **3** or **4** is determined by the solvent used. This uncharacterized precursor does not exist for the other phosphines, or the formation of a cluster similar to **4** is not possible from this precursor.

The close structural relationship between compounds **1** and **3** is also demonstrated by their UV/Vis spectra (Figure 4). Compound **1** displays a significant absorption band at 471 nm, with shoulders at 452 nm, 493 nm, 540 nm, 587 nm, and 637 nm (orange). Likewise, **3** presents a strong absorption band at 482 nm with several smaller bands and shoulders (blue). The UV/Vis spectra of the three different Au_32_ clusters are almost the same, since the influence of the ligand on the electronic situation of the nucleus is small. The absorption of **1** appears analogous to that of compound **3**; however, blue shifted by 10 nm. Such small shifts in similar clusters were also observed for the [Au_25_(SCH_2_Ph)_18_]^-^ cluster which revealed minimal differences in the UV/Vis spectra when compared with both the neutral species and doped cluster compounds, such as Au_24_Pt or Au_24_Hg [17].

To investigate the impact of the sterics of the phosphine ligand on the cluster, we performed DFT calculations based on the experimentally determined structures in the solid state, using the BP86 functional and the def-SV(P) basis set. The gold–gold distances in the DFT-optimized structure exceeded the experimentally measured distances by 12.5 pm for the inner shell and 13.5 pm for the outer shell. Within **1**, a total of 24 valence electrons are expected (one electron from every gold atom in oxidation state 0). Taking the Jellium model [35] into account, 24 is not a magic number for a closed-shell system. The closest Jellium number is 20, which resembles a 1S^2^1P^6^1D^10^2S^2^ electronic configuration.

Therefore, we expect a partly filled F state. This situation is prone to Jahn–Teller distortion, as is also the case for the Au_32_ clusters. A similar situation was realized for **1**, as the three energetically highest occupied orbitals, HOMO, HOMO-1, and HOMO-2, show an F-type character (Figure 5a). The remaining four F orbitals are the unoccupied orbitals LUMO, LUMO+1, LUMO+2, and LUMO+4 (Figure 5b).

This Jahn–Teller distortion is energetically favorable, because it is not possible to evenly fill seven orbitals with six electrons and is visible in the icosahedral core of **1**, which is not perfect but tilted, leading thus to a splitting of the F orbitals. Additionally, six of the seven F orbitals are energetically more favorable than the 2S orbital, which is represented by the LUMO+3 orbital (Figure 5c). The electronic configuration, with respect to the Jellium model, can be described as 1S^2^1P^6^1D^10^1F^6^. A comparable arrangement is observed for **3**, with the exception that LUMO+4 represents the 2S orbital. This similarity is not surprising, since the only variation, apart from the phosphine ligands, are the two AuCl units, which thus has no influence on the number of valence electrons.

To further investigate the steric influence of the phosphine ligands, we checked the following Reactions (1) and (2) using DFT calculations.
Au_30_(P*^i^*Pr_2_*^n^*Bu)_12_Cl_6_
**1** + 2 *^i^*Pr_2_*^n^*BuPAuCl **2** → 2 *^i^*Pr_2_*^n^*BuP + Au_32_(P*^i^*Pr_2_*^n^*Bu)_12_Cl_8_(1)
Au_30_(P*^n^*Bu_3_)_12_Cl_6_ + 2 *^n^*Bu_3_PAuCl → 2 *^n^*Bu_3_P + Au_32_(P*^n^*Bu_3_)_12_Cl_8_
**3**(2)

The DFT calculations show that with *^i^*Pr_2_*^n^*BuP as the ligand, **1** is favored by 120.3 kJ/mol over a Au_32_(P*^i^*Pr_2_*^n^*Bu)_12_Cl_8_ cluster. This indicates that the addition of the two AuCl units is sterically hindered so that no place is available for the chlorine substituents. This argument is further supported by the fact that the P*^n^*Bu_3_-stabilized Au_30_ cluster is disfavored compared with **3** by 43.4 kJ/mol. 

These calculations confirm that the phosphine ligand influences the amount of gold atoms in the cluster. Next to the examples, in which the phosphine determines the form of the cluster during the reduction, there are also examples whereby an additional phosphine changes the form of a gold cluster. The addition of the bulky bis(diphenylphosphino)propane to a Au_9_(C_6_H_4_-*p*-Me)_8_(NO_3_)_3_ cluster yields the smaller Au_6_(dppp)_4_(NO_3_)_2_ cluster. The transformation results in a drastic geometric change, as the Au_9_ cluster, which can be described by a centered icosahedron from which an equatorial rectangle has been removed, is transformed into the Au_6_ cluster with the shape of a tetrahedron in which two opposite edges are capped by a gold atom [36]. This Au_6_ cluster can also be synthesized from the tetrahedral Au_4_(PPh_3_)_4_I_2_ cluster once dppp has been added.

To check the influence of the steric bulk of the phosphine ligands on the stability of a Au_32_ or Au_30_ cluster, experiments were conducted on the growth and degradation of compounds **1** and **3** with the addition of *^n^*Bu_3_P and *^n^*Bu_3_PAuCl to **1** and *^i^*Pr_2_*^n^*BuP to **3**. These reactions were studied using ^31^P NMR. To make sure that all phosphine ligands are exchanged, the phosphine was added in a large stoichiometric overdose. Both reactions resulted in the vanishing of signals for the cluster compounds (Appendix A), while the signal for the unbound phosphine was detected, which stabilized the initial cluster. This clearly indicates that, indeed, a substitution of the phosphine ligand took place. Therefore, in the case of Au_32_(P*^n^*Bu_3_)_12_Cl_8_
**3**, the substitution of *^n^*Bu_3_P by *^i^*Pr_2_*^n^*BuP should lead to a less stable Au_32_ cluster that might decompose. This is supported by the UV/Vis spectrum of the reaction solution, which looks more like a spectrum from gold nanoparticles. In contrast to this, the UV/Vis spectrum of **1** after the addition of *^n^*Bu_3_P exhibited similarities to the spectra of **1** and **3**, with minor variations (Appendix A). After adding (*^n^*Bu_3_P)AuCl to the assumed Au_30_ cluster with *^n^*Bu_3_P ligands, the ^31^P spectrum displayed a signal at 62.1 ppm (Appendix A) that was similar to the signal of **3** at 62.8 ppm. This suggests that the conversion of **1** to **3** (Equation (3)) was successful, since no similar signal was detected in the reactants. It should be noted that this transformation was not successful with *^n^*Pr_3_P.
Au_30_(P*^i^*Pr_2_*^n^*Bu)_12_Cl_6_
**1** + 12 *^n^*Bu_3_P → 12 *^i^*Pr_2_*^n^*BuP + Au_30_(P*^n^*Bu_3_)_12_Cl_6_
Au_30_(P*^n^*Bu_3_)_12_Cl_8_ + 2 (*^n^*Bu_3_P)AuCl → Au_32_(P*^n^*Bu_3_)_12_Cl_8_
**3**(3)

## 3. Materials and Methods

Unless otherwise stated, all commercial reagents were used as received. The reactions were carried out in an inert gas atmosphere using standard Schlenk techniques. Sodium was used to pre-dry the THF and cyclopentane; NaK for THFd_8_, hexane, and heptane; CaH_2_ for pentane; and 3 Å molecular sieves for CDCl_3_ and C_6_D_6_. CDCl_3_ and C_6_D_6_ were dried with 3 Å molecular sieves. *^i^*Pr_2_*^n^*BuP and **3** were synthesized as described earlier by our group [25,37].

In addition, **2** was synthesized in situ by reacting 321 mg (1 mmol) (THT)AuCl, suspended in 20 mL THF and 0.2 mL (1 mmol) *^i^*Pr_2_*^n^*BuP. The solution became colorless, and the solvent was removed after one hour, resulting in a white solid.

^1^H NMR (CDCl_3_) δ: 0.94 ppm (3H, t), 1.16–1.31 ppm (12H, m), 1.39–1.53 ppm (2H, m), 1.53–1.8 ppm (4H, m), and 2.06–2.25 ppm (2H, m); ^31^P NMR (CDCl_3_) δ: 52,6 ppm (1P, s)

Moreover, **1** was synthesized by reducing 406 mg (1 mmol) of **2**, which was suspended in 20 mL ethanol, with 38 mg (1 mmol) NaBH_4_. The suspension quickly turned black, and after one hour, the solvent was removed via vacuo. The resulting brown solid was extracted using 20 mL of cyclopentane, filtered, and stored at 6 °C. In case there was any oil present, the extract was filtered again. After several weeks, **1** crystallized in the form of a black rhombus. (Yield after several crystallizations: 4 mg (1.46%)).

^31^P NMR (C_6_D_6_) δ: 81.39 (6P, s) and 85.69 (6P, s)

The crystallization of **1** is also possible in pentane, hexane, and heptane instead of cyclopentane. However, crystallization with these solvents has to be performed at −30 °C, and the crystals are not suitable for SCXRD.

The NMR spectra were recorded on AVIIIHD-300 ((*^i^*Pr_2_*^n^*BuP)AuCl, (*^n^*Bu_3_P)AuCl, **1** + *^n^*BuP_3_, **1** + (*^n^*Bu_3_P)AuCl, and **1** + (*^n^*Pr_3_P)AuCl) and Bruker AvanceII+400 (*^n^*Bu_3_P, *^i^*Pr_2_*^n^*BuP, and **3** + *^i^*Pr_2_*^n^*BuP) spectrometers in CDCl_3_, C_6_D_6_, and THF-d_8_. The chemical shifts are given in ppm against the external standards SiMe_4_ (^1^H) and 85% phosphoric acid (^31^P). 

The UV/Vis measurements were carried out using the PG INSTRUMENTS LIMITED T60 spectrophotometer under inert conditions. The background spectra were taken with identical cuvettes and solvents.

The X-ray crystal analysis was performed as follows: Crystals were mounted on the diffractometer at 100 K. The data were collected on a Bruker APEX II DUO diffractometer equipped with an IμS microfocus sealed tube and QUAZAR optics for monochromated MoKα radiation (λ = 0.71073 Å) and equipped with an Oxford Cryosystems cryostat. A semiempirical absorption correction was applied using the program SADABS. The structure was solved with direct methods and refined against F2 for all observed reflections. Programs used: SHELXS and SHELXL within the Olex2 program package [38,39,40]. CCDC 2312885 (**2**) and 2312886 (**1**) contain the supplementary crystallographic data for this paper.

All quantum chemical calculations were performed at a DFT level with the BP86 functional (exchange: LDA + Becke (B88); correlation: LDA (VWN) + Perdew (P86)), using turbomole with the Tmolex GUI. The def-SV(P) basis was set, and additionally for the gold atoms, the def-ecp was used [41,42,43,44,45,46].

## 4. Conclusions

By studying the influence of the steric bulk of the phosphine ligand on the core of a metalloid gold cluster, we synthesized the novel Au_30_(P*^i^*Pr_2_*^n^*Bu)_12_Cl_6_ cluster from the reduction of (*^i^*Pr_2_*^n^*BuP)AuCl with NaBH_4_ and characterized it by x-ray crystallography, UV/Vis, and NMR spectroscopy. Au_30_(P*^i^*Pr_2_*^n^*Bu)_12_Cl_6_ is closely related to the three Au_32_ clusters, which can be synthesized with the sterically less demanding alkyl-substituted phosphines, but is missing two AuCl units. The UV/Vis analysis confirmed the resemblance in structure, as exhibited by a similar spectrum, with a blue shift of 10 nm for the Au_30_ cluster. The DFT calculations indicate that the steric impact of the phosphine ligand is decisive for the creation of either a Au_32_ or a Au_30_ cluster. For the P*^i^*Pr_2_*^n^*Bu ligand, the preference of Au_30_ formation over Au_32_ is 120.4 kJ/mol. The conversion from **3** to **1** resulted in Au nanoparticles; however, the NMR and UV/Vis indicate a conversion from **1** to **3** after the addition of *^n^*Bu_3_P and (*^n^*Bu_3_P)AuCl. This raises the question of whether heteroatoms could be added instead of AuCl units, thereby doping the Au_32_ cluster. Further research is necessary to investigate whether it is possible to remove more AuCl units to synthesize a Au_30-x_Cl_6-x_(PR_3_)_12_ (x = 1, 2, 3…) cluster with other, even more sterically demanding, phosphine ligands or to discover a cluster similar to Au_20_.

## Figures and Tables

**Figure 1 molecules-29-00286-f001:**
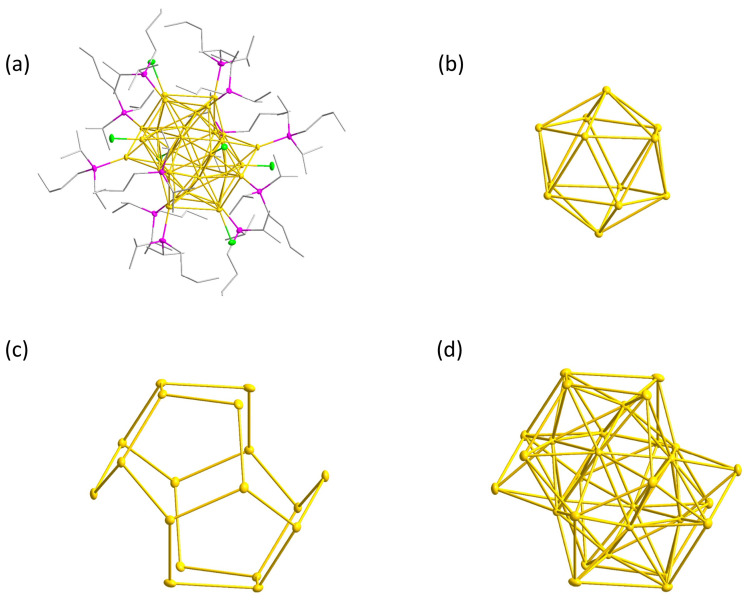
(**a**) Molecular structure of 1 in the solid state. All atoms, except for carbon, are displayed as thermal ellipsoids with a 50% probability. Hydrogen atoms are omitted for clarity. Au: yellow; P: pink; Cl: green. (**b**) The Au_12_ hollow icosahedral inner shell of **1**. (**c**) The Au_18_ pentagonal dodecahedral outer shell of **1**. (**d**) The Au_30_ core of **1**, composed of the Au_12_ inner shell and the Au_18_ outer shell.

**Figure 2 molecules-29-00286-f002:**
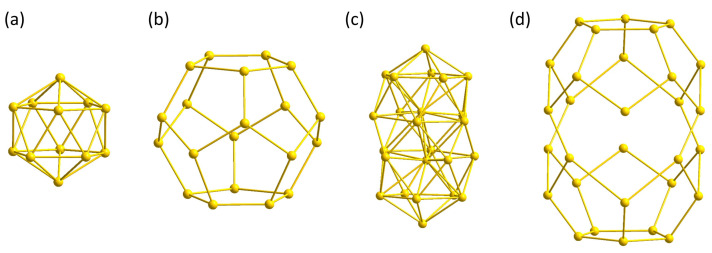
(**a**) Inner icosahedral Au_12_ shell of **3**; (**b**) outer pentagon dodecahedral Au_20_ shell of **3**; (**c**) inner biicosahedral Au_24_ shell of **4**; (**d**) outer bipentagon dodecahedral Au_30_ shell of **4**.

**Figure 3 molecules-29-00286-f003:**
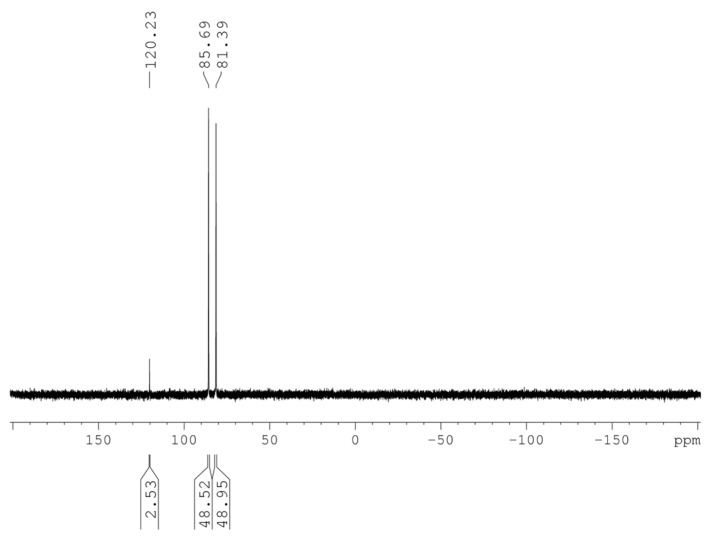
^31^P NMR spectrum of 1. The signals are at 85.69 and 81.38 ppm with similar intensities. The signal at 120.22 is most likely from a Au32 cluster.

**Figure 4 molecules-29-00286-f004:**
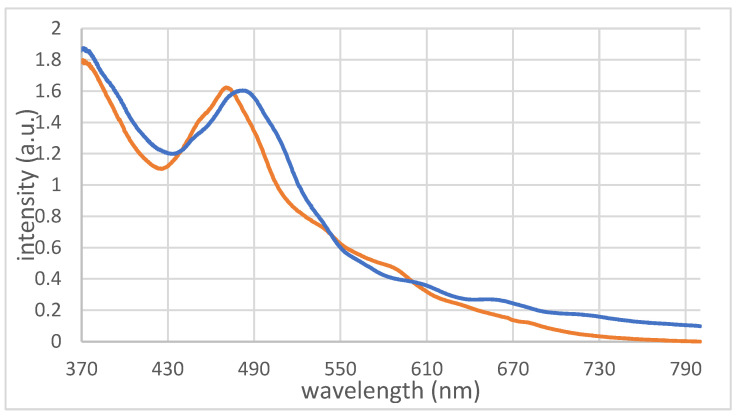
UV/Vis spectra of **1** (orange) and **3** (blue) in benzene.

**Figure 5 molecules-29-00286-f005:**
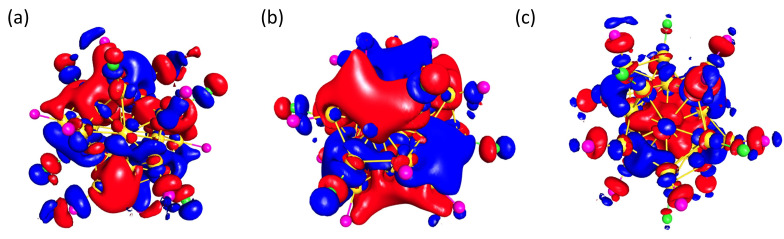
Visualization of the (**a**) HOMO [0.01], (**b**) LUMO [0.01], and (**c**) LUMO+3 [0.015] of **1**.

**Table 1 molecules-29-00286-t001:** Comparison of the bond lengths in **1**, **3**, and **4** in pm.

Bond/Distance	Au_30_(P^i^Pr_2_^n^Bu)_12_Cl_6_ 1	Au_32_(P^n^Bu_3_)_12_Cl_8_ 3	Au_54_(PEt_3_)_18_Cl_12_ 4
Au-Au (inner shell)	284.9 ± 5.7	285.1 ± 3.3	284.8 ± 10.1
Au-Au (outer shell)	314.8 ± 17.3	312.6 ± 11.7	306.0 ± 12.5
Au-Au (between shells)	278.4 ± 5.7	276.8 ± 3.2	281.5 ± 9.4

## Data Availability

Data are contained within the article and Appendix A.

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
