# Peer review of "Au30(PiPr2nBu)12Cl6—An Open Cluster Provides Insight into the Influence of the Sterical Demand of the Phosphine Ligand in the Formation of Metalloid Gold Clusters"

_molecules, 2024, doi:10.3390/molecules29020286_

Round 1

Reviewer 1 Report

Comments and Suggestions for Authors

Markus Strienz and Andreas Schnepf have reported the Au30(PiPr2nBu)12Cl6 with crystal structures, which is interesting and important. 31P NMR spectrum was applied to study the surface phosphine ligand. The paper is good written. I suggest the publication of the work on Molecules after some changes.

 Line 243: The Au30(PiPr2nBu)12Cl6 1 + 2 iPr2nBuPAuCl 2 → iPr2nBuP + Au32(PiPr2nBu)12Cl8 should be Au30(PiPr2nBu)12Cl6 1 + 2 iPr2nBuPAuCl 2 → 2 iPr2nBuP + Au32(PiPr2nBu)12Cl8.

The influence of the phosphine ligand in the synthesis of metalloid gold clusters is well studied. But how about the influence of the phosphine ligand on the physical property of the metalloid gold clusters, for example, the FL property? Some of work on ligand effect should be cited, e.g. 10.1021/acs.jpclett.3c00509, etc.

 The Au30 clusters can transfer to Au32 in the presence of RPAuCl complex. Can the Au32 clusters transfer to Au30 under other conditions.

Author Response

Markus Strienz and Andreas Schnepf have reported the Au30(PiPr2nBu)12Cl6 with crystal structures, which is interesting and important. 31P NMR spectrum was applied to study the surface phosphine ligand. The paper is good written. I suggest the publication of the work on Molecules after some changes.

Answer: We thank the referee for his positive assessment of our work and changed it with respect to his comments as follows:

  • Line 243: The Au30(PiPr2nBu)12Cl6 1 + 2 iPr2nBuPAuCl 2 → iPr2nBuP + Au32(PiPr2nBu)12Cl8 should be Au30(PiPr2nBu)12Cl6 1 + 2 iPr2nBuPAuCl 2 → 2 iPr2nBuP + Au32(PiPr2nBu)12Cl8.

We changed the reaction equation and thank the reviewer.

  • But how about the influence of the phosphine ligand on the physical property of the metalloid gold clusters, for example, the FL property? Some of work on ligand effect should be cited, e.g. 1021/acs.jpclett.3c00509, etc.

Most of the physical properties such as absorption are determined by the cluster core. The most important factor of the ligand is that it determines the shape and size of the cluster core. To measure a direct influence of the ligand on the cluster, we would need a second Au30 cluster with a different phosphine ligand, which wasn't possible to synthesize. If this were possible, we would be able to measure, for example, the conductivity as a direct function of the ligand, as was done for the Au32 clusters. However, we agree with the reviewer that additional information on the role of ligands on the physical properties of the cluster could provide a better understanding and added examples to the manuscript where the direct influence of ligands on solubility, luminescence and absorption were studied.

  • The Au30 clusters can transfer to Au32 in the presence of RPAuCl complex. Can the Au32 clusters transfer to Au30 under other conditions.

We tried to change the phosphine ligand of the Au32 clusters by adding the iPr2nBuP in a large excess, to exchange the ligands to the cluster and, after the ligands are exchanged, to dissociate a R3PAuCl unit. Instead, both the Au32nBu and the Au32nPr cluster decomposed after several hours for the nPr3P stabilized cluster and after several days for the nBu3P stabilized cluster Further experiments were not performed because we didn't find a soft way to remove two gold atoms from a Au32 cluster, without destroying it.

Reviewer 2 Report

Comments and Suggestions for Authors

The paper by Strienz and Schnepf reports the synthesis and characterization of a series of ligands-protected Au clusters. The experimental part is adequately described, the crystal structures have been analyzed by standard XRD and are complemented by NMR and UV-vis spectroscopic data in solution. However the presentation of the theoretical results is rather poor, and important methodological details are missing that hamper a full judgement of the work. Here are my concerns:

1) which software has been used for the geometry optimization of the nano clusters?

2)explain the choice of the particular exchange-correlation (xc)  potential used and basis set. These details are crucial for a qualitatively correct analysis of the results. The authors could try to reproduce by time-dependent DFT at least the low-energy portion of the UV-Vis spectra and compare with the experiment. The comparison can be used as a check that the computed electronic structure of the clusters is at least reasonable.

3) Are these systems closed-shell or open-shell systems? In the latter case, is spin-polarization used or calculations use positively-charged clusters?

4) How do the optimized structures (I suppose that the geometry optimization has been conducted in vacuo, otherwise please specify) compare with the solid-state experimental structures?

Unless these  aspects are clarified I cannot recommend the article to be published in Molecules.

Author Response

The paper by Strienz and Schnepf reports the synthesis and characterization of a series of ligands-protected Au clusters. The experimental part is adequately described, the crystal structures have been analyzed by standard XRD and are complemented by NMR and UV-vis spectroscopic data in solution. However, the presentation of the theoretical results is rather poor, and important methodological details are missing that hamper a full judgement of the work. Here are my concerns:

Answer: We thank the reviewer for his hints and suggestions and have taken his concerns into account as follows:

  • Which software has been used for the geometry optimization of the nano clusters?

We used the turbomole software in the version 7.4.1 with the Tmolex GUI in the version 4.4.1.

  • Explain the choice of the particular exchange-correlation (xc)  potential used and basis set. These details are crucial for a qualitatively correct analysis of the results. 

The used xc potential and basis set is the standard method in our group, because it provides realistic values with moderate computational times (which is important for systems of this size). Comparisons with other methods in the past have shown that the increased accuracy is not worth the higher computational cost (this is especially the case for calculations based on the crystal structure, when the results can be compared with experimental data).

  • The authors could try to reproduce by time-dependent DFT at least the low-energy portion of the UV-Vis spectra and compare with the experiment.

TDDFT calculations were performed, but the calculated spectra showed less similarity to the experimental spectra. This is a common occurrence for gold cluster compounds of this size, and therefore we have chosen not to discuss it further.

  • Are these systems closed-shell or open-shell systems? In the latter case, is spin-polarization used or calculations use positively-charged clusters?

The compound is assumed to be a closed shell system due to the presence of 24 electrons. Additionally, the location of the NMR signals are consistent with other closed shell clusters, further supporting this conclusion.

  • How do the optimized structures (I suppose that the geometry optimization has been conducted in vacuo, otherwise please specify) compare with the solid-state experimental structures?

The calculations were performed in vacuum. The calculated geometry of the cluster is identical to the experimental structure. There is a small difference in the bond lengths, which we have added to the manuscript.

Reviewer 3 Report

Comments and Suggestions for Authors

The present manuscript deals with the preparation of a new type of open (non-superatomic) molecular Au30 cluster stabilized with phosphine ligands of well calibrated steric bulk. The cluster has been crystallographically characterized and its structural data have been extensively compared with related clusters previously reported in the literature. Finally, a computational study has been carried out to elucidate the electronic structure of the cluster.

Phosphine-stabilized molecular gold clusters have a long history and several examples with different structural characteristics have been reported in recent years. Nevertheless, the work reported herein has been competently done and indeed represents a case in which the steric bulk of the ligand has been purposely optimized to achieve the synthetic result. On the basis of the above, I can recommend publication in Molecules, provided the authors revise their manuscript according to the points listed below.

-          My main concern with this manuscript is the employed synthetic methodology. According to the authors, crystals of the title cluster are obtained in very low yield from solution after several weeks of standing. Is this preparation method reproducible? Can the authors claim that the title cluster is indeed present in solution right after completion of the reduction? Some evidence of cluster formation by e.g. HRMS (the authors report that they were unable to characterize their cluster by “mass spectrometry” but do not report which spectrometric technique they tried out) or maybe even 31P NMR spectroscopy would be welcome. Otherwise, there is the possibility that the cluster is serendipitously formed during the crystallization procedure.

-          I do not quite understand while the authors define in several places the oxidation state of some gold atoms in the cluster as “+/-0”. Why “+/-“? Of course, there is an average oxidation state for the gold atoms which is between 0 and 1 but this “uncertainity” regards both the 0 and the +1 oxidation state.

-          Throughout the text, monophosphine gold(I) educts are better written without brackets, e.g. R3PAuCl

-          Page 7, equation 3: I guess the intermediate Au30 cluster species should have 6 chlorides, not 8.

Author Response

The present manuscript deals with the preparation of a new type of open (non-superatomic) molecular Au30 cluster stabilized with phosphine ligands of well calibrated steric bulk. The cluster has been crystallographically characterized and its structural data have been extensively compared with related clusters previously reported in the literature. Finally, a computational study has been carried out to elucidate the electronic structure of the cluster. Phosphine-stabilized molecular gold clusters have a long history and several examples with different structural characteristics have been reported in recent years. Nevertheless, the work reported herein has been competently done and indeed represents a case in which the steric bulk of the ligand has been purposely optimized to achieve the synthetic result. On the basis of the above, I can recommend publication in Molecules, provided the authors revise their manuscript according to the points listed below.

Answer: We thank the reviewer for the overall positive assessment of our work and we revised it according to the comments of the referee as listed below:

  • According to the authors, crystals of the title cluster are obtained in very low yield from solution after several weeks of standing. Is this preparation method reproducible?

The reviewer is right that the yield of the cluster is low. However, it's important to note that the formation of the cluster requires a large number of bonds to be broken and formed, so even though there are examples of clusters with larger amounts, a low yield is not unusual in cluster chemistry. Yes, the method is reproducible and has been done multiple times.

  • Can the authors claim that the title cluster is indeed present in solution right after completion of the reduction?

No, we cannot. Instead, we assume that the cluster is formed later, since there are no signals in the 31P NMR during the synthesis as well as after extraction with pentane, and this would also explain the long time it takes for the cluster to crystallize, since recrystallization is possible in a few days.

  • I do not quite understand while the authors define in several places the oxidation state of some gold atoms in the cluster as “+/-0”. Why “+/-“? Of course, there is an average oxidation state for the gold atoms which is between 0 and 1 but this “uncertainity” regards both the 0 and the +1 oxidation state.

We do this to make clear, that this is a zero and not a O, but we agree, that this could be confusing and changed it in the manuscript.

  • Throughout the text, monophosphine gold(I) educts are better written without brackets, e.g. R3PAuCl

Since the iPr2nBuP ligand looks written confusing, we decided to use backets to make it more clear.

  • Page 7, equation 3: I guess the intermediate Au30 cluster species should have 6 chlorides, not 8.

We agree with the reviewer and have thankfully changed this.

Round 2

Reviewer 2 Report

Comments and Suggestions for Authors

I am satisfied with the amendments on the manuscript and with the authors' answers to the issues I had raised in my previous report. I think this work  can now be accepted as is.